international students; suicide prevention; gatekeeper training; safeTALK; cultural adaptations

**Corresponding author:**
Samuel McKay;
Email: sam.mckay@unimelb.edu.au

# Suicide prevention for international students: A single-arm mixed methods evaluation of the LivingWorks safeTALK program in Australia

Christina Ng[1], Michelle Lamblin[1,2], Jo Robinson[1,2] and Samuel McKay[1,2] (ID)

[1]Orygen, 35 Poplar Road, Parkville, VIC, 3052, Australia and [2]Centre for Youth Mental Health, The University of Melbourne, Parkville, VIC, Australia

## Abstract

International students frequently report suicidal thoughts and behaviours, but often do not seek help. We evaluated the feasibility, acceptability, and preliminary effectiveness of an adapted version of safeTALK suicide prevention training for international students. Eight workshops were delivered in Melbourne, Australia (N = 128; 62.5% female, M age = 23.4). In this single-arm study, surveys were completed pre-, post-, and three months post-training, and 17 participants completed follow-up interviews. The training was rated as acceptable, helpful, and safe. Linear mixed models indicated increased confidence to intervene and stronger intentions to refer individuals to formal help sources, with improvements sustained at follow-up. Suicide stigma showed a small post-training reduction that was not sustained. Suicide literacy only improved three months post-training. Attrition limited inferences about long-term effects. Qualitative feedback supported the training's value but highlighted the need for further cultural adaptation. Findings support adapted gatekeeper training as a promising strategy for suicide prevention among international students.

## Impact statement

International students are a growing part of global education systems, yet they face disproportionately high risks of suicide and significant barriers to accessing mental health support. These challenges are amplified by cultural stigma, communication differences, and limited awareness of available services. Despite this, there are currently no widely used, evidence-based suicide prevention programs designed specifically for international students.

This study is the first to adapt and evaluate the safeTALK suicide prevention training for international students in a real-world setting. The findings demonstrate that an adapted gatekeeper training model is not only acceptable and engaging but also develops critical skills, including confidence, suicide literacy, and readiness to intervene. Importantly, the training helped international students feel empowered to support their peers and more confident navigating mental health issues, both personally and within their communities.

By focusing on equipping students themselves as "gatekeepers," the program bypasses many of the traditional barriers to help-seeking, such as stigma and unfamiliarity with services. The findings also demonstrate that even light-touch cultural adaptations, like using culturally relevant examples and refining language, can significantly improve engagement and applicability.

The results have clear implications for tertiary institutions globally. Universities can implement adapted gatekeeper training to proactively build safer, more supportive student communities, especially for diverse cohorts. This study provides a replicable model for culturally responsive prevention efforts that can be expanded to other high-risk groups.

## Introduction

International students in tertiary education experience higher rates of suicidal behaviour compared to their domestic peers (Veresova et al., 2024). Stressors such as cultural adjustment, financial pressures, language barriers, and discrimination contribute to international students' poor mental health and suicidality (Khanal and Gaulee, 2019; Veresova et al., 2024; Clough et al., 2019). A reluctance to seek mental health support can further heighten their vulnerability (Clough et al., 2019). Despite this, no evidence-based suicide prevention programs are specifically designed for international students (McKay et al., 2023).

Suicide prevention approaches within education settings are typically categorised into three levels: universal, selective, and indicated (Robinson et al., 2018). Universal interventions target entire populations, such as school-wide education campaigns, selective interventions focus on at-risk subgroups, and indicated interventions address individuals exhibiting suicidal

behaviours. Selective and indicated interventions can reduce suicidal thoughts and behaviours (Robinson et al., 2018). However, these types of interventions often rely on individuals actively seeking help – an area where international students face barriers due to stigma, cultural norms, and service access issues (e.g., cost: Clough et al., 2019; McKay et al., 2023). Universal suicide prevention training programs can overcome these barriers by equipping community members to identify and support individuals at risk without the need for active help-seeking. International student cohorts represent an ideal avenue for such universal prevention initiatives, enabling all members of a higher-risk community to receive training regardless of individual vulnerability (Reifels et al., 2021; McKay et al., 2023).

Meta-analyses indicate that suicide prevention training can prevent suicide attempts and deaths in education settings (Pistone et al., 2019; Walsh et al., 2022). Training young people as "gatekeepers" enables them to support their peers while enhancing their own mental health skills and knowledge (Pistone et al., 2019). One such intervention, Suicide Alertness for Everyone (safeTALK) training, equips community members to identify individuals at risk of suicide, understand contributing factors, and connect them with support (Burnette et al., 2015; LivingWorks Australia, 2025). safeTALK improves suicide-related knowledge, confidence, and help-seeking behaviours, including among young people in education settings (Mellanby et al., 2010; Bailey et al., 2017) and has no iatrogenic effects (Wilson and Neufeld, 2017). Although safeTALK has been successfully adapted for diverse groups (North Western Melbourne Primary Health Network, 2022; Mueller-Williams et al., 2023), it has not been tested with international students. Given that international students experience unique barriers to help-seeking, including cultural stigma around suicide, indirect communication norms, and lower familiarity with local mental health resources (Clough et al., 2019; McKay et al., 2023), research is needed to determine its acceptability and effectiveness.

## Current study

This study evaluated the appropriateness, acceptability, effectiveness, and real-world application of an adapted safeTALK training for international students. We assessed if the training was engaging, relevant, and culturally suitable (appropriateness and acceptability); improved suicide literacy, suicide stigma, helping self-efficacy for suicide crises, and intentions to encourage help-seeking for suicidal individuals (effectiveness); and whether participants intended to or applied the skills or knowledge in real-world situations (Application of training). The research questions were:

1. Is the training appropriate for, and acceptable to, international students?
2. Does the training change international students' suicide prevention skills, knowledge, or stigma?
3. To what extent do participants apply the skills and knowledge gained from the training?

## Method

### Study design

This study used a longitudinal mixed methods design. Recruitment was conducted through six education institutions and one student accommodation provider. Participants provided electronic consent through a REDCap form and were assessed at three time points:

Time One (T1) before training; Time 2 (T2) immediately after training; and Time 3 (T3) three months post-training. Those who completed T2 were invited to complete T3. Participants who completed T1-T3 were entered into a random draw to win one of five $100 vouchers. At T2, participants could indicate their interest in an interview and were compensated $30 for participating.

### Intervention

LivingWorks safeTALK is a face-to-face, four-hour suicide skills alertness gatekeeper training workshop for anyone aged 15 or above. The program aims to increase trainees' alertness to suicide and their ability to connect those thinking of suicide to further help. Eight workshops (Attendee M = 15.63, range = 9–32 participants) were delivered at six educational institutions and one student accommodation provider. The program was adapted for international students through a consultation workshop. After completing the standard training, participants reviewed each program component (videos, TALK steps, role-play scenarios, trainer delivery) to identify barriers and facilitators to engagement and skill use, including cultural concerns such as stigma, norms around discussing suicide, and help-seeking preferences (See McKay et al., 2024 for a more detailed summary of the adaptation process). Adaptations emphasised the prevalence of suicidality in this population, incorporating relevant data and contextually specific examples of life stressors (e.g., academic failure, cultural shame, family and parental conflicts, racism, social status changes, and language difficulties). Barriers to discussing suicide, such as stigma, fears of burdening family, religious beliefs, language challenges, negative service experiences, and visa concerns, were addressed. The program incorporated international student-specific support services and relevant role-play scenarios while retaining core safeTALK components (e.g., steps to ask about suicide, discussions, role-play activities, and video content) to maintain fidelity and key learning objectives. All adaptations were developed in consultation with a LivingWorks Trainer, and training delivery followed the standard facilitator manual to ensure alignment with the original program structure and competencies.

### Survey measures

#### Demographic information (T1 only)

Participants indicated their age, gender, nationality, time spent living in Australia, time spent living in other countries, education level, field of study, most commonly spoken language, native language, training motivations and how they learned about the training.

#### RQ1: appropriateness and acceptability

**Satisfaction with safeTALK (T2 only).** Acceptability and satisfaction with the safeTALK training were assessed using a questionnaire originally developed for a school-based trial (Byrne et al., 2022). The questionnaire included four items evaluating whether participants found the training "enjoyable," "upsetting," and "worthwhile," each rated on a 3-point scale (1 = not at all, 3 = very), as well as whether they would recommend the training to others (yes/no). Four additional items were developed in consultation with culturally diverse research staff to assess the cultural acceptability of the training. These items evaluated whether participants felt the training adequately prepared them to provide initial support, considered their cultural background and learning preferences, and was relevant and applicable to their cultural context (yes/no).

**Training Content Helpfulness (T2 only).** A purpose-designed questionnaire assessed the helpfulness of content areas, including suicide content, attitudes and beliefs about suicide, group simulations,

role plays, and the focus on creating a "suicide-safer community." Five items were rated on a 10-point scale (1 = Not Helpful to 10 = Very Helpful), with higher scores indicating greater perceived helpfulness.

**General Feedback Question (T2 and T3).** A single open question asked if participants had further feedback about their experience of the program and/or survey.

### RQ2: effectiveness

**Intentions to encourage help-seeking (T1–3).** The General Help-Seeking Questionnaire (GHSQ) assessed participants' intentions to encourage formal and informal help-seeking for someone at risk of suicide (Wilson et al., 2005). It consists of 13 items rated on a 7-point scale (1 = Extremely Unlikely to 7 = Extremely Likely), summed separately for formal and informal sources. Higher scores indicate greater intentions to encourage help-seeking from each source. The formal (α = .90) and informal help-seeking subscales (α = .82) demonstrated good internal consistency.

**Suicide Literacy (T1–3).** Suicide literacy was assessed using the Literacy of Suicide Scale – Short Form (LOSS-SF; Batterham et al., 2013), which contains 12 items addressing common knowledge about suicide. Items were presented in a *true/false* format. Correct responses received a score of 1, while incorrect responses were scored 0, yielding a total score between 0 and 12.

**Suicide Stigma (T1–3).** Suicide stigma was assessed using the Stigma of Suicide Scale - Short Form (SOSS-SF), which includes 16 items rated on a 5-point Likert scale ranging from 1 (Strongly Disagree) to 5 (Strongly Agree; Batterham et al., 2013). There are three subscales: stigma (8 items), isolation (4 items), and glorification (4 items). Scores were calculated by summing item responses, with only the stigma subscale reported in this study. Higher scores reflect greater suicide stigma. The stigma subscale demonstrated excellent reliability (α = .93).

**Helping Self-Efficacy for Suicide Crises (T1–3).** Participants' perceived self-efficacy to engage in activities to prevent or assist others in managing a suicide crisis was measured using an adapted version of the self-efficacy scale originally developed for parents of suicidal youth (Czyz et al., 2018). The adapted version included nine items scored on a 10-point Likert scale ranging from 1 (Not Confident) to 10 (Very Confident). Items are summed, and higher scores indicate greater self-efficacy. This measure demonstrated excellent reliability (Cronbach's α = .94).

### RQ3: application of training

**Intentions to use skills learned in the training (T2 only).** Behavioural intent to use the skills from the safeTALK training was measured using five questions asking how much participants intended to use the training and specific skills in their community on a 7-point scale ranging from 1 (Very Unlikely) to 7 (Extremely Unlikely).

**Indication, Referral and Skill Usage (T3 only).** A purpose-designed questionnaire reflecting the core expected competencies for safeTALK was developed for this study. It contained 8 items assessing whether participants had used the skills they learned in the training, how they had used them and the services they had referred to in the past 3 months. Questions were also asked about the outcomes of skill use or, if not used, the reasons for non-use.

### Semi-structured interview guide

A semi-structured interview guide (see supplementary materials) captured participants' perspectives on the training, including its cultural acceptability, whether they gained and applied new skills or knowledge, and suggestions for improvements.

### Analytic approach

Quantitative data were analysed using RStudio (Posit team, 2025). Linear Mixed Models (LMM) were applied to examine the changes in the key outcome measures, including suicide literacy, suicide stigma, suicide helping self-efficacy and help-seeking intentions, over time. This model was chosen due to its ability to account for both fixed effects (e.g., time) and random effects (e.g., individual variability between participants; Hoffman, 2015).

A reflexive thematic analysis of the interview data was conducted using NVivo, following the six steps outlined by Braun and Clarke (2022). The analysis followed an interpretivist-constructivist paradigm, enabling deep exploration of students' experiences, attitudes, and opinions while acknowledging the researcher's reflexivity. Interview transcripts were reviewed for familiarisation and analysed inductively, with codes refined to prioritise student experiences. The codes were organised into themes, which were reviewed and refined to ensure accurate data representation and coherence in writing. Consistent with reflexive thematic analysis, inter-rater reliability was not calculated. Instead, rigour was supported through multiple readings, iterative coding, and reflexive discussions between CN, who conducted all phases of the analysis and SM, who provided guidance and critical review of developing themes.

## Results

### Participants

Participants were 128 international students aged 18 or above and enrolled in tertiary education in Victoria, Australia (See Table 1 for an overview of the sample demographics). A total of 126 students completed T1, 105 completed T2 (2 of which did not complete T1), and 40 students completed T3. The sample was predominantly female (62.5%). The students represented diverse nationalities, with the most common being Chinese (33.6%), Vietnamese (11.7%), and Indian (10.9%). Most students were enrolled in an undergraduate (32.8%) or postgraduate (32.0%) program and reported a native language other than English (93.6%), although over half (54.7%) spoke English most of the time. Most students (94.5%) had no prior experience with suicide prevention training. Sensitivity analyses comparing baseline demographics of T3 completers (n = 40) and non-completers (n = 88) indicated no significant differences on any variable except course level, with completers more likely to be undergraduates (50.0% vs. 25.0%) and less likely to be enrolled in diploma/certificate programs (5.0% vs. 30.7%).

A subsample of 17 participants participated in an interview on Microsoft Teams conducted by CN with support from SM. Interviews lasted between 10 and 33 min (M = 17.58, SD = 7.74) and were audio-recorded via Microsoft Teams, automatically transcribed, and then manually reviewed. Sensitivity analyses comparing interviewees and non-interviewees identified no significant differences in age, gender, education level, language, or previous training, but interviewees were more likely to be undergraduates (58.8% vs. 28.8%) and less likely to be enrolled in diploma/certificate programs (0% vs. 22.7%).

### Mixed methods reporting approach

The main findings are organised by research question, with quantitative results presented first, followed by qualitative insights. The qualitative analysis identified six overarching themes: 1) Learning Together: An Engaging and Enjoyable Experience, 2) Accessibility and Practicality of Training Materials, 3) Cultural Relevance and

**Table 1.** Participant demographics

|  | All participants | | Interview subsample | |
|---|---|---|---|---|
| **Age (Mean, SD)** | 23.40 | (5.46) | 24.88 | (6.80) |
| *Gender* | | | | |
| Male | 44 | (34.4%) | 6 | (35.3%) |
| Female | 80 | (62.5%) | 10 | (58.8%) |
| Non-binary | 1 | (0.8%) | 1 | (5.9%) |
| Gender nonconforming | 1 | (0.8%) | - | - |
| *Years in Australia* (Mean, SD) | 1.65 | (3.03) | 2.35 | (3.67) |
| *Nationality* | | | | |
| China | 43 | (33.6%) | 2 | (11.8%) |
| Vietnam | 15 | (11.7%) | 4 | (23.5%) |
| India | 14 | (10.9%) | 3 | (17.6%) |
| Taiwan | 5 | (3.9%) | 2 | (11.8%) |
| Philippines | 5 | (3.9%) | 2 | (11.8%) |
| Bangladesh | 4 | (3.1%) | - | - |
| Colombia | 4 | (3.1%) | - | - |
| Singapore | 4 | (3.1%) | - | - |
| Malaysia | 3 | (2.3%) | - | - |
| Indonesia | 3 | (2.3%) | - | - |
| Thailand | 3 | (2.3%) | - | - |
| Korea | 3 | (2.3%) | - | - |
| Brazil | 2 | (1.6%) | - | - |
| Iran | 2 | (1.6%) | - | - |
| Japan | 2 | (1.6%) | - | - |
| Mauritius | 2 | (1.6%) | 1 | (5.9%) |
| Burma | 1 | (0.8%) | - | - |
| Cambodia | 1 | (0.8%) | - | - |
| Germany | 1 | (0.8%) | 1 | (5.9%) |
| Greece | 1 | (0.8%) | - | - |
| Kenya | 1 | (0.8%) | - | - |
| Mexico | 1 | (0.8%) | - | - |
| Myanmar | 1 | (0.8%) | - | - |
| Nepal | 1 | (0.8%) | - | - |
| Nigeria | 1 | (0.8%) | - | - |
| Persia | 1 | (0.8%) | 1 | (5.9%) |
| Sri Lanka | 1 | (0.8%) | 1 | (5.9%) |
| Turkey | 1 | (0.8%) | - | - |
| *Previous Education* | | | | |
| Secondary School | 38 | (29.7%) | 7 | (41.2%) |
| Certificate/Diploma | 22 | (17.2%) | 2 | (11.8%) |
| Undergraduate Degree | 51 | (39.8%) | 7 | (41.2%) |
| Postgraduate Degree | 14 | (10.9%) | 1 | (5.9%) |
| Other | 1 | (0.8%) | - | - |

*(Continued)*

**Table 1.** *(Continued)*

|  | All participants | | Interview subsample | |
|---|---|---|---|---|
| *Current Education* | | | | |
| Certificate/Diploma | 29 | (22.7%) | - | - |
| Undergraduate Degree | 42 | (32.8%) | 10 | (58.8%) |
| Postgraduate Degree | 41 | (32.0%) | 5 | (29.4%) |
| Other | 14 | (10.9%) | 2 | (11.8%) |
| *Native Language*[a] | | | | |
| English | 8 | (6.3%) | 2 | (11.8%) |
| Other | 117 | (91.4%) | 15 | (88.2%) |
| *Main language Spoken* | | | | |
| English | 70 | (54.7%) | 11 | (64.7%) |
| Other | 56 | (43.8%) | 6 | (35.3%) |
| *Previous Suicide Prevention Training* | | | | |
| Yes | 5 | (3.9%) | 1 | (5.9%) |
| No | 121 | (94.5%) | 16 | (94.1%) |

*Note:* N = 128 for the full sample; however, only 126 participants completed the first survey, resulting in 1.6% missing data on descriptive variables.
[a]n = 125 due to missing data for this variable. Interview subsample: N = 17.

Communication Styles, 4) Helping Others: Awareness and Support, 5) Readiness and Application of Skills, and 6) Empowerment and Collective Responsibility. These themes offer deeper insight into students' experiences, complementing the statistical findings by illustrating how and why the training was effective, along with areas for improvement.

## RQ1: appropriateness and acceptability

### General appropriateness and acceptability

Most participants reported being "satisfied" or "very satisfied" with the training (83.8%). Nearly all participants found the training enjoyable (99.0%), worthwhile (98.1%), and not upsetting (98.1%). Additionally, 99.0% would recommend the training to others and felt it gave them the confidence to provide initial support to someone experiencing suicidal thoughts.

Participants rated the training highly ($\geq 8$ on a 1–10 scale), with most finding the content helpful (89.5%, M = 8.90, SD = 1.24) and valuing discussions on suicide attitudes and beliefs (85.7%, M = 8.82, SD = 1.34). Group simulations, including asking about suicide (72.4%, M = 8.45, SD = 1.73) and role plays (75.2%, M = 8.39, SD = 1.83), were well-rated. Similar ratings were received for the focus on creating a suicide-safer community (85.7%, M = 8.77, SD = 1.28). These high satisfaction ratings were echoed in qualitative interviews, where participants described the training as engaging, well-structured, and culturally relevant.

**Theme 1 - Learning Together: An Engaging and Enjoyable Experience.** Participants described safeTALK as engaging and enjoyable, attributing this to its interactive format and skilled facilitation. They found the content easy to understand and appreciated the facilitator's ability to make it relatable. Group discussions around cultural triggers and barriers to discussing suicide were particularly valued, offering a space to share personal experiences, challenge misconceptions, and explore different cultural perspectives

on mental health: "The thing I enjoyed the most was probably the discussions… it gave us an insight into… what we already know, but also some of the stuff we thought we knew but wasn't actually true." (Participant 10).

Students enjoyed forming bonds with others and learning about cultural similarities and differences. This allowed them to gain a more nuanced perspective and find commonalities. The role-playing exercises were another highlight, not just because they allowed students to practice "hands-on" skills but also because of their interactivity: "The little role play… was like the best part for me" (Participant 17). These exercises enabled participants to apply what they had learned, supporting knowledge consolidation. This positive engagement was consistent with the strong quantitative ratings for group simulations and role plays.

**Theme 2 – Accessibility and Practicality of Training Materials.** Interview participants felt the workshop was well-structured, appreciating the balance between presentation and interactive discussion. They appreciated the variety in the training materials, which enhanced engagement, "I love the fact they have used like different video clips to enhance the message." (Student 9). The timing of the training sessions was also crucial. While sessions were offered throughout the semester, students preferred having the training at the beginning when they were still facing transitional challenges. This preference for early delivery aligns with participants' ratings of the training's relevance to their current circumstances.

Participants valued the physical materials provided, such as the handbook and reminder card, which reinforced the key approaches taught in the training. However, some students found the handbook less practical outside the workshop and suggested digital versions in multiple languages for better accessibility, "I have the card with me, but … if I had a digital version… that would be helpful for the future" (Student 8).

### Cultural appropriateness and acceptability

Most students felt that the training was designed with consideration for different cultural backgrounds (86.7%). They also believed the training was consistent with their cultural learning styles and preferences (97.1%) and found it relevant and applicable to their cultural context (86.7%). However, students provided written feedback suggesting that it would be helpful to provide tailored approaches to conversations about mental health and suicide, including less direct phrasing, building rapport, and incorporating non-English examples to address diverse cultural norms and taboos. For instance, a student from Vietnam noted, "It's very awkward to talk directly about mental health since it's like a taboo topic…it could be done better by letting them feel comfortable with you first."

**Theme 3 - cultural relevance and communication styles.** Participants found the language respectful and accessible, appreciating that example scenarios were broadly applicable. However, some students found the direct approach challenging and suggested adapting communication styles to align with cultural norms. As one student reflected, "The approach … is a little bit direct in my opinion because, in my culture, we don't come up to people and say, do you think of suicide?" They went on, "After having several conversations, they feel that I'm a trustworthy person. Rather than seeing them… in trouble and asking them directly." (Participant 11).

While the training demonstration videos and role-playing scenarios were helpful, they did not always resonate with interview participants' specific experiences, who suggested including more culturally relevant scenarios to enhance relatability: "Maybe the video examples? Like they are not like bad or anything. I just feel like they're a little bit cliché., I came from a very sheltered

South East Asian background. We don't talk about suicide like that" (Participant 14).

This theme emphasises the importance of cultural alignment in training content. It also intersects with effectiveness, as stigma and cultural barriers may limit the application of skills post-training and reduce reach among international students. These cultural factors may explain why some participants who had opportunities to use the skills at T3 (see next section) chose not to intervene, highlighting the need for culturally tailored approaches to ensure training is applied in real-world settings.

### RQ2: effectiveness

*Intentions to encourage help-seeking, suicide literacy, suicide stigma, and self-efficacy to help someone thinking of suicide*
The LMM analyses showed that intentions to encourage formal help-seeking increased significantly from T1 to T2, with these improvements largely retained at T3. In contrast, there was no significant change in intentions to encourage informal help-seeking from T1 to T2, although scores were significantly lower at T3. Suicide literacy did not increase between T1 and T2 but showed a significant improvement by T3. A decrease in was observed in suicide stigma from T1 to T2, but this was not sustained at T3. Helping self-efficacy increased substantially from T1 to T2 and decreased slightly from T2 to T3, although T3 levels remained significantly higher than at baseline. A summary of the results is presented in Table 2. These observed changes in literacy, stigma, and self-efficacy were reflected in qualitative themes that described greater awareness, confidence, and interpersonal sensitivity.

**Theme 4 - helping others: awareness and support.** Participants said the safeTALK training enhanced their mental health literacy and awareness of available resources, with some surprised by the number of support options and reassured they were not alone: "I learned that there is more help available than I expected… when they showed us the slide with all the helplines available" (Participant 14).

The training also helped normalise conversations about mental health and suicide, making students feel more confident to support others and seek help. One student mentioned, "safeTALK training let me be more sensitive to others' emotions… I think I've become more proactive in caring about others and caring for myself." (Participant 4).

Students felt that the training improved their interpersonal skills, reporting being more present and engaged in conversations, particularly when interacting with friends and peers: "The skills I've learnt, I think it definitely helped me in terms of being more engaged and aware in conversations when it comes to talking to friends and peers" (Participant 10).

### RQ3: application of training

*Intentions to use skills learned in the training*
At T2, most participants indicated (76.2%, M = 6.23, SD = 0.91) they intended to use the skills they had learned (rated as ≥6 on a scale from 1 to 7) particularly for raising the question of suicide (73.3%, M = 6.01, SD = 1.13), seeking more information about a plan (76.2%, M = 6.19, SD = 0.91), encouraging help-seeking (86.7%, M = 6.49, SD =0.77), calling a crisis line (81.0%, M = 6.25, SD = 0.98), and accompanying someone to get help (82.9%, M = 6.23, SD = 1.02).

*Actual usage of skills learned in the training*
At T3, 62.5% (N = 25) of participants reported using the skills gained from the safeTALK training, while 37.5% (N = 15) indicated

**Table 2.** Raw means and standard deviations along with pairwise comparisons for help-seeking intentions, suicide literacy, suicide stigma, and helping self-efficacy across timepoints

| | T1 | T2 | T3 | Comparison | Mean difference | Cohen's d |
|---|---|---|---|---|---|---|
| Formal help-seeking | 27.96 (6.16) | 30.98 (5.30) | 29.98 (4.48) | T1 − T2 | 2.99 [1.86, 4.13]*** | 0.70 [0.43, 0.96] |
| | | | | T1 − T3 | 2.31 [0.71, 3.91]* | 0.54 [0.16, 0.92] |
| | | | | T2 − T3 | −0.68 [−2.30, 0.94] | −0.16 [−0.54, 0.22] |
| Informal help-seeking | 20.78 (4.95) | 21.84 (4.81) | 19.18 (5.53) | T1 − T2 | 1.03 [−0.11, 2.16] | 0.24 [−0.03, 0.50] |
| | | | | T1 − T3 | −1.31 [−2.92, 0.29] | −0.31 [−0.68, 0.07] |
| | | | | T2 − T3 | −2.34 [−3.97, −0.71]* | −0.55 [−0.93, −0.16] |
| Suicide literacy | 8.84 (1.87) | 9.21 (1.66) | 10.12 (1.43) | T1 − T2 | 0.24 [−0.10, 0.58] | 0.19 [−0.08, 0.46] |
| | | | | T1 − T3 | 0.92 [0.43, 1.42]** | 0.73 [0.33, 1.12] |
| | | | | T2 − T3 | 0.68 [0.18, 1.18]* | 0.54 [0.14, 0.93] |
| Suicide stigma | 17.23 (7.48) | 15.67 (7.57) | 14.50 (7.43) | T1 − T2 | −1.33 [−2.58, −0.08]* | −0.29 [−0.56, −0.02] |
| | | | | T1 − T3 | −1.19 [−3.15, 0.47] | −0.29 [−0.69, 0.10] |
| | | | | T2 − T3 | −0.01 [−1.83, 1.81] | 0.00 [−0.40, 0.40] |
| Helping self-efficacy | 56.62 (14.32) | 76.16 (12.28) | 69.47 (14.67) | T1 − T2 | 19.93 [16.99, 22.87]*** | 1.80 [1.49, 2.11] |
| | | | | T1 − T3 | 13.61 [9.39, 17.83]*** | 1.23 [0.83, 1.63] |
| | | | | T2 − T3 | −6.32 [−10.65, −1.99]* | −0.57 [−0.96, −0.18] |

*Note*: Values in parentheses represent standard deviations (SD). Values in square brackets represent 95% confidence intervals (CI). Cohen's d represents the standardised mean difference.
*p < .05. ** p < .01 *** p < .001.

they had not. Among those who had used the skills, 76.0% (N = 19) reported identifying invitations or signs that someone might be suicidal, and 16.0% (N = 4) stated that individuals at risk had been referred to them. Conversely, participants who had not used the skills, 86.7% (N = 13) stated that they had not interacted with someone at risk of suicide. One participant encountered someone at risk but lacked the confidence to intervene. The relatively low rate of direct crisis interventions at T3 aligns with qualitative accounts, where many described using the skills in general supportive contexts rather than suicide-specific situations. These narratives suggest that, although readiness was high, opportunities for application were rare.

**Theme 5: readiness and application of skills.** Participants generally felt that the training had equipped them with the knowledge and confidence to support others, even if most had not yet encountered an acute suicide crisis. Several described applying their skills in preventive or supportive contexts, such as recognising signs of distress, listening without judgement, and offering information about help services. For example, one student recounted approaching a stranger in their accommodation after noticing changes in mood:

> I felt like they might be struggling… I started a friendly conversation and they opened up… I told them if they needed to talk or get help… they can use some resources or come to me. They reacted pretty positive… they liked that these places exist (Participant 1)

The role-play component was consistently credited with preparing participants for real-world situations. One student who supported a friend overseas said practising scenarios helped them "be open in that specific scenario" and recall what to say and what not to say, which "actually helped her a lot" (Participant 12). Some adapted their approach for particular contexts. For example, one person working with teenagers found they needed to "water it down and talk to them more on their level… you can't directly ask if they feel suicidal" (Participant 13). Others noted cultural norms that may limit disclosure, with one explaining that friends "might not want to tell someone… their inner world" (Participant 2). These accounts suggest that safeTALK fostered a readiness to act and an increased awareness of distress cues, but that sustained application may rely on opportunities to practise and on adapting the skills to different relational and cultural contexts.

**Theme 6: empowerment and collective responsibility.** Interview participants with lived experience of suicide and mental ill-health found the training empowering and reassuring, appreciating the commitment of others to support those in crisis: "It also helps that knowing that people who are not experiencing these thoughts themselves are also trying to help us" (Student 14). Attending alongside other international students fostered a sense of belonging and collective responsibility. "I think just the feeling of being involved or just included, like as an international student… as a collective… I think that is really good" (Student 12). Some were inspired to share their skills and knowledge with their home communities, such as giving talks in schools to help young people "be more open about their emotions… and [make it] less of an awkward feeling" (Student 12).

## Discussion

This study is the first to preliminarily evaluate an adapted version of safeTALK for international students. The training was well-received and culturally acceptable, with interactive elements, such as discussions on cultural barriers and role-plays, particularly valued. While cultural elements were rated highly, qualitative feedback highlighted the need for further adaptation, such as simplified language and more culturally relevant scenarios. The training improved self-efficacy and intentions to encourage formal help-seeking, with many participants expressing readiness to apply these skills. By the three-month follow-up, over half of the respondents had applied the skills. However, most interviewees had not

encountered a suicide crisis, instead applying them in general support contexts. Suicide literacy improved only at the three-month follow-up, and stigma was reduced immediately post-training, but this decrease was not maintained. Overall, the adapted safeTALK appears acceptable and effective for improving suicide prevention competencies among international students, though further cultural adaptation may enhance reach, resonance, and sustained use.

The findings align with previous evaluations of safeTALK in diverse and community samples in improving knowledge, self-efficacy and intentions to intervene (Bailey et al., 2017; Holmes et al., 2023; Mueller-Williams et al., 2023). Outcomes likely stem from the program's structured yet interactive format, with cultural adaptations enhancing rather than determining its effectiveness. Participants valued role-plays, suicide attitudes discussions, and the focus on fostering a suicide-safer community, consistent with prior research on participatory learning for suicide prevention (Cross et al., 2011). The training also appeared to shift preferences toward formal help-seeking, mirroring findings from other gatekeeper programs with international students (Kiran et al., 2023) and fostering collective responsibility and community-mindedness (Cross et al., 2011; Holmes et al., 2021). Notably, suicide literacy gains emerged only three months post-training, coinciding with reports that participants had applied their skills by follow-up. This suggests that real-world application and reflection may be critical for consolidating knowledge, underscoring the value of longitudinal follow-up in gatekeeper training evaluations.

Consistent with previous research on gatekeeper training that reports mixed effects on stigma (Grosselli et al., 2024; Liu et al., 2025), safeTALK only led to a temporary reduction in suicide stigma. This short-lived impact may be due to the program's brevity, its emphasis on skill-building over attitude change, and the deeply embedded nature of stigma in some cultures. Given stigma's role as a barrier to help-seeking (McCullock and Scrivano, 2023), future safeTALK adaptations could integrate more lived-experience content. While safeTALK currently uses scripted videos with actors, research suggests that authentic lived experience stories, particularly those reflecting diverse cultural backgrounds, are more effective at reducing stigma than other educational content (Patten et al., 2012; McCullock and Scrivano, 2023; Crockett et al., 2025). It may also be beneficial to explore stigma measures that better align with the training focus. The measure used in this study (SOSS) assesses attitudes toward people who have died by suicide, whereas safeTALK centres on recognising and responding to suicidal ideation and behaviour. Measures capturing help-seeking stigma (self-stigma and perceived public stigma; Crockett et al., 2025), social distance toward people with suicidal thoughts, and comfort asking directly about suicide may be more sensitive to training-related attitudinal shifts.

Cultural alignment remains a challenge. Students from East Asian backgrounds often rely on indirect expressions of distress, which Western-focused direct questioning may not suit (Chu et al., 2020). Without adaptation, this mismatch may reduce confidence and effectiveness (McKay and Meza, 2024). Culturally adapted QPR training in Guyana addressed similar issues by incorporating local risk factors, behavioural cues, and culturally appropriate rapport-building techniques (Persaud et al., 2019), offering a potential model to enhance safeTALK. While cultural expressions of distress vary, international students also share common stressors such as acculturation difficulties, social isolation, and challenges navigating mental health systems (Veresova et al., 2024). Rather than tailoring solely to specific cultural groups, adaptations should use flexible, nuanced approaches that address both varied communication styles

and these shared experiences (McKay and Meza, 2024). For safeTALK, this could include role-plays reflecting international student realities, direct and indirect communication strategies, and culturally relevant examples of help-seeking. Peer-led delivery may further enhance accessibility and engagement, provided facilitators receive adequate supervision and debriefing (Gillard and Holley, 2014; Muehlenkamp and Quinn-Lee, 2023; Wexler et al., 2017).

The impact of suicide prevention training is shaped not only by cultural fit but also by local context. This study took place in Victoria, Australia, which hosts the country's second-largest number of international students, with Indian students most represented compared with Chinese students nationally (Department of Education, 2025). Local factors such as work rights legislation, shifting government policy, and high reported discrimination may influence suicidal ideation and engagement with supports (McKay et al., 2023; Veresova et al., 2024). These supports are often poorly adapted for international students in Western nations (Sanci et al., 2022; Sakız and Jencius, 2024) and may be even less accessible in other countries (e.g., China; Wu et al., 2021), limiting trainees' willingness or ability to refer peers. Contextual differences can substantially affect training outcomes; for example, Chinese students in Malaysia face distinct language barriers, cultural and religious differences, and social integration challenges (Xue and Kaur Mehar Singh, 2025). Such factors may shape how students respond to training and their likelihood of referring peers, underscoring the need for research across diverse settings.

## Limitations

First, qualitative and quantitative feedback on cultural appropriateness diverged, potentially due to social desirability bias. Alternative methods, such as behavioural measures, could capture more nuanced views. Second, attrition was very high (40 participants at T3 vs. 105 at T2), limiting statistical power to detect sustained effects (e.g., stigma) and precision in estimating skill use at follow-up. Attrition may also have introduced response bias, as undergraduates were more likely than diploma students to complete T3 and the interviews. Future studies should trial enhanced retention strategies such as incentives or varied follow-up formats. Third, females were overrepresented in the training cohort, limiting representativeness for males. Fourth, several measures (e.g., *Satisfaction with safeTALK*, *Training Content Helpfulness*, and *Indication, Referral and Skill Usage*) were purpose-designed or adapted for this study. While they demonstrated acceptable internal consistency, they lack full psychometric validation, warranting further reliability and validity testing in larger samples. Finally, as this was a single-arm study without a comparator group, changes observed cannot be attributed solely to program participation.

## Conclusion

This study provides preliminary evidence that adapted safeTALK training is acceptable and effective in improving knowledge, confidence, and intentions to intervene and refer peers to formal help. However, suicide literacy gains only occurred at the three-month follow-up, stigma reductions were temporary, and high attrition limits conclusions about long-term impact. Qualitative feedback underscored the need for further cultural adaptation. Future studies should explore culturally aligned content, peer-delivery models, and enhanced retention strategies. Despite these limitations, findings suggest gatekeeper training holds promise as a suicide prevention

approach for international students, warranting further investigation into its long-term effectiveness.

**Open peer review.** To view the open peer review materials for this article, please visit http://doi.org/10.1017/gmh.2025.10082.

**Supplementary material.** The supplementary material for this article can be found at http://doi.org/10.1017/gmh.2025.10082.

**Data availability statement.** The data that support the findings of this study are available from the corresponding author, SM, upon reasonable request.

**Acknowledgements.** The authors would like to thank the international students who participated and LivingWorks Australia for their support of this research.

**Author contribution.** Christina Ng: Data curation, Formal analysis, Visualisation, Writing—original draft, Writing—review and editing. Michelle Lamblin: Conceptualization, Methodology, Writing—review and editing. Jo Robinson: Conceptualization, Methodology, Resources, Writing—review and editing. Samuel McKay: Conceptualization, Funding acquisition, Methodology, Data curation, Formal analysis, Project administration, Supervision, Writing—review and editing.

**Financial support.** This work was funded by a grant from the Victorian Government Department of Jobs, Skills, Industry and Regions. Jo Robinson is funded by a National Health and Medical Research Council Investigator Grant (ID2008460) and a Dame Kate Campbell Fellowship from the University of Melbourne.

**Competing interests.** The authors declare no conflicts of interest.

**Ethics approval.** This study was approved by the University of Melbourne Human Research Ethics Committee (#27010). All participants provided written informed consent before enrolment in the study, which was conducted in accordance with the Declaration of Helsinki.

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
