## [Reviewer Report]

Please refer to the submitted review for minor amendments to consider before final publication.

Review

Suicide Prevention for International Students: A Mixed Methods Evaluation of the LivingWorks safeTALK Program in Australia

The study reported in this paper evaluated the appropriateness, acceptability, effectiveness, and real-world application of an adapted safeTALK training in suicide prevention for international students. The study addressed three research questions:

1. Is the training appropriate for, and acceptable to, international students?

2. Does the training change international students' suicide prevention skills, knowledge, or stigma?

3. To what extent do participants apply the skills and knowledge gained from the training?

Quantitative data were collected at three time points:

Time One (T1) before training; demographic information only

Time 2 (T2) immediately after training focused on RQ1

Time 3 (T3) three months post-training. Those who completed T2 were invited to complete T3. At T2, participants could indicate their interest in an interview.

Intervention

LivingWorks safeTALK is a face-to-face, four-hour suicide skills alertness gatekeeper training workshop for anyone aged 15 or above. The program aims to increase trainees’ alertness to suicide and their ability to connect those thinking of suicide to further help.

Program was adapted for international students through a consultation workshop:

• Adaptations emphasised the prevalence of suicidality in this population, incorporating relevant data and contextually specific examples of life stressors.

• Barriers to discussing suicide, such as stigma, fears of burdening family, religious beliefs, language challenges, negative service experiences, and visa concerns were addressed.

• Incorporated international student-specific support services and relevant role-play scenarios while retaining core safeTALK components (steps to ask about suicide, discussions, role-play activities, and video content)

General comments

This paper reports on a training intervention for suicide prevention among international students in one Australian state. The opening paragraph argues that international students are an at-risk group, suggesting that the intervention described falls into the selective category. However, the same paragraph concludes with the proposition that “International student cohorts represent an ideal avenue for such universal prevention initiatives”, of which Suicide Alertness for Everyone (safeTALK) training is one. It is a minor quibble, rather than a major issue, but it would be good to remove the problem in the published version.

That said, the study was well-designed, in that the intervention employed a training package previously successfully adapted for diverse groups and modified it further for appropriateness, acceptability, effectiveness, and real-world application of safeTALK training for international students.

Intervention

LivingWorks safeTALK is a face-to-face, four-hour suicide skills alertness gatekeeper training workshop for anyone aged 15 or above. The program aims to increase trainees’ alertness to suicide and their ability to connect those thinking of suicide to further help.

The program had previously been used successfully with diverse groups, it was adapted through a consultation workshop to improve the fit for international students:

• Adaptations emphasised the prevalence of suicidality in this population, incorporating relevant data and contextually specific examples of life stressors.

• Barriers to discussing suicide, such as stigma, fears of burdening family, religious beliefs, language challenges, negative service experiences, and visa concerns were addressed.

• Incorporated international student-specific support services and relevant role-play scenarios while retaining core safeTALK components (steps to ask about suicide, discussions, role-play activities, and video content).

Recruitment for the training was conducted through six education institutions and one student accommodation provider. A total of 126 students completed T1, 105 completed T2, and 40 students completed T3, three months post-training. Considering that T3 addressed the extent to which participants applied the skills and knowledge gained from the training, the relatively low participation in this phase of the research limits the strength of these findings. Seventeen participants (64.7% female, age M = 24.88 & SD = 6.80, from a range of countries including Vietnam, India, China, Taiwan, and Philippines), offering a diversity of cross-cultural perspectives.

Sampling and participant attrition

Participants were aged 18+ and recruited across six education institutions and one student accommodation provider, indicating a wide pool of potential participants. Some discrepancies in the demographic information reported are unaccounted for, presumably because participants did not complete all sections of the demographic instrument. Of the 94.8% who disclosed their gender identity, 60.2% were female, 33.1% were male and 1.50% identified as Other. It is not clear whether the predominance of female participants is reflective of gender distribution in the international student population or whether females were more interested in participation in the program. Given that the suicide intervention training described in the paper is designed to equip community members with the skills to identify individuals at risk of suicide, further consideration might be given to gender as an efficacy factor in the role that “gatekeepers” might be able to fulfil.

As a longitudinal study concluding three months after initial recruitment of participants, the strength of its findings suffered from sample attrition, as the researchers acknowledged in their discussion of limitations. Although demographic data identified gender and educational levels at the point of recruitment, education levels were not considered in successive cycles of data collection. Initially there were approximately equal numbers of undergraduate (31.6%) and postgraduate (30.8%) participants compared to 21.8% Diploma or certificate enrolees, leaving 15.8% of the sample unaccounted for. Of the initial 126 participants, only 40 remained in T3, three months after the training program. Seventeen participants (64.7% female, approximating the initial gender representation) self-selected from the 105 who completed T2 participated in the interview. It might be illuminating to do further analysis of the cohort who completed stage 3 and the interviews, particularly in relation to efficacy, effectiveness and application of the program.

Findings

The interactive elements of the intervention – discussions, role-play, video clips and group simulations were highly rated. Although the intervention was generally rated highly for cultural appropriateness and acceptability, interview data offered a more nuanced view of this theme in which participants expressed a view that some of the interventions were too direct and felt too “confrontational” for direct application in their cultural milieu. This information is potentially useful for further adaptations of the package, even if only by increasing the opportunities for participants to discuss ways in which variations to accommodate culturally diverse sensitivities in engagement strategies.

Two emergent themes (Theme 4 - Helping Others: Awareness and Support; Theme 6: Empowerment and Collective Responsibility) suggested wider social effects than suicide prevention which might enhance the well-being of international students more generally, ameliorating the sense of isolation experienced by many international students.

Conclusion

As the researchers acknowledged, there are limitations to this study, as the first to address suicidal tendencies among international students in Australia. Despite the limitations, this study does offer opportunities for application, replication and adaptation through further research, policy development and practice. As Australia continues to rely on international students to underpin the budgets of educational institutions, it is timely that this research offers some directions for taking up responsibility for developing sounder policies and practices for their support. Recruitment of international students is not merely an Australian phenomenon; this research has potential to contribute more widely to research policy and practice to support international student well-being.

---

## [Reviewer Report]

I am glad I have reviewed “Suicide Prevention for International Students: A Mixed Methods Evaluation of the LivingWorks safeTALK Program in Australia”. This is a timely and important study. This manuscript presents a mixed-methods evaluation of an adapted safeTALK suicide prevention training program for international students in Australia. While the topic is highly relevant and the study addresses a critical gap in the literature, there are several shortcomings. My review will highlight these areas for improvement, urging the authors to consider these points for future research and revisions.

1. The manuscript generally adheres to a clear and concise academic writing style. The language is accessible, and the flow of information is logical, particularly in the introduction and methods sections. However, there are instances where the presentation could be more succinct, especially in describing certain qualitative findings, which sometimes feel repetitive or overly descriptive rather than analytical. The integration of quantitative and qualitative results, while present, could be more seamless, potentially through a more explicit narrative connecting the two data streams rather than simply presenting them sequentially.

2. The theoretical underpinning of the study, which implicitly relies on gatekeeper training models for suicide prevention, is sound. The manuscript effectively highlights the elevated suicide risk among international students and the barriers they face in help-seeking, thus establishing a clear rationale for the intervention. The conceptualization of “gatekeepers” among international students to overcome traditional help-seeking barriers is well-articulated. However, the paper could benefit from a more explicit theoretical framework that guides the adaptations made to safeTALK and explains why these specific cultural adaptations were expected to be effective, rather than simply describing them.

3. The study employs a longitudinal mixed-methods design, which is a strength, allowing for both quantitative assessment of effectiveness and qualitative insights into acceptability and real-world application.

• The recruitment through educational institutions and student accommodation providers is appropriate. However, a significant limitation is the high attrition rate from T2 (immediately post-training) to T3 (three months post-training), with only 40 out of 105 T2 participants completing T3 surveys. This substantial drop-off severely limits the conclusions that can be drawn about the long-term effectiveness of the intervention. While the authors acknowledge this limitation, its impact on the robustness of long-term findings is critical.

• The use of validated scales like the General Help-Seeking Questionnaire (GHSQ) and the Literacy of Suicide Scale - Short Form (LOSS-SF) is a strength. However, the purpose-designed questionnaires for “Satisfaction with SafeTALK,” “Training Content Helpfulness,” and “Indication, Referral and Skill Usage” could benefit from further psychometric validation beyond internal consistency, particularly given their central role in assessing appropriateness, acceptability, and application.

• The description of the safeTALK adaptation process is good, noting the incorporation of culturally specific stressors and barriers. However, further detail on how fidelity to the core safeTALK components was maintained while introducing adaptations would strengthen the methodology.

• The semi-structured interviews and reflexive thematic analysis are appropriate for gaining in-depth understanding. The inter-rater reliability or verification of themes could be more explicitly detailed to enhance confidence in the qualitative findings, especially as one author conducted all phases of the analysis with guidance from another.

4. The study claims to be the first to adapt and evaluate safeTALK for international students in a real-world setting. This indeed represents a novel contribution to the field of suicide prevention, particularly for a high-risk and often underserved population. The findings that an adapted gatekeeper training model is acceptable, engaging, and develops critical skills like confidence and suicide literacy are important initial steps. However, the primary novelty lies in the adaptation and preliminary evaluation rather than a definitive demonstration of long-term effectiveness, which is hampered by the aforementioned attrition issues.

5. The presentation of quantitative results is clear, using linear mixed models to analyze changes over time, and a summary table is provided. The qualitative findings are presented thematically, providing rich contextual depth to the quantitative results. The use of participant quotes effectively illustrates the themes. However, the depth of findings, particularly regarding the application of training in real-world situations, is limited. While intentions to use skills were measured , the actual reported usage and outcomes from T3 are not as thoroughly explored or presented, which is crucial for a program aiming for real-world impact.

6. The discussion section appropriately interprets the findings, linking them back to the research questions and existing literature. It acknowledges the study’s limitations, particularly the attrition and the need for further cultural adaptation. The implications for tertiary institutions are also clearly stated. However, the discussion could be significantly strengthened by a more critical self-reflection on the limitations and their implications for the generalizability and impact of the findings. Specifically, the lack of change in suicide stigma should be discussed in greater depth, as it remains a significant barrier for international students.

7. For revisions, the authors are strongly urged to integrate insights from recent and highly relevant literature on inclusive mental health support and suicidal ideation among international students. This would enrich the theoretical discussion, inform further cultural adaptations, and strengthen the methodological design to address persisting challenges:

• https://doi.org/10.1017/gmh.2024.1. This paper could provide a framework for understanding and implementing inclusive delivery components, directly addressing the cultural adaptation aspects highlighted as needing further attention in the current manuscript.

• https://doi.org/10.1177/00110000211002458. This study offers a deeper dive into the cultural, academic, and interpersonal factors contributing to suicidal ideation, which could inform more nuanced adaptations of suicide prevention programs and target specific risk factors beyond general gatekeeper training.

• https://doi.org/10.1007/s10447-023-09540-1. This work on structural components of counseling services could provide feedback into systemic changes that complement individual-level gatekeeper training, offering a more holistic approach to suicide prevention in this population. It also directly relates to the broader institutional implications discussed.

---

## [Editor Report]

Dear Dr McKay and colleagues,

Thank you for your submission to the journal. This is a thoughtful and well-written piece, and an important contribution to the literature. In addition to the reviews by our expert reviewers, I have a quick point:

While Victoria represents a dense part of Australia, it would be good to further contextualise the difference in international student population in the location of the study vs. Australia in general. Furthermore, international students, while often facing similar challenges, are quite heterogeneous across the world. The manuscript’s discussion could benefit from a more in-depth exploration of how international students in Australia, while similar to international students globally, are also unique. An example of a very different cohort would be Indonesian students residing in Malaysia. 

Please carefully read the reviews and respond to them point by point. I look forward to your resubmisson.

Thank you and all the best,

Dr Sandersan Onie

---

## [Reviewer Report]

I appreciate the authors' dedicated efforts to revise and improve the articles. The final editing will be important, as some editable minor language issues remain.

---

## [Editor Report]

Dear Dr McKay,

Thank you for taking the time to make substantial revisions to the manuscript. I am suggesting some further edits:

1. On page 24, implicit measures are suggested. The literature on implicit measures since 2012 have been controversial, and I would not suggest that you include that statement. 

2. An important finding is that SafeTALK repeatedly does not reduce stigma - which may be an even bigger issue in international populations, whether in Australia or abroad. I would suggest expanding on a discussion on why this might be, and what the program could do to address this; or perhaps if we have not been measuring it well. 

3. Note that on page 16, you say that there was an observed change in stigma, whereas on page 15, there was no significant change in suicide stigma. 

4. I would like to see a discussion on why suicide literacy could have increased only at T3, given that it did not increase following training. 

I think overall, this is important work - but our understanding and ability to address suicide and train for help provision in international settings is quite nascent, even for programs with many publications such as SafeTALK. As such, beyond the points above, a nuanced discussion on why the data appear the way they do, and what might help, would help the field understand how to best prevent suicide in these settings - lest as a field we grow complacent in improving our programs. 

Thank you and all the best,

Dr Sandersan Onie

---

## [Editor Report]

Thank you to the authors for their resubmission, which has addressed the comments put forth. I have some further comments which need to be addressed prior to publication:

1. In the abstract, please rephrase ‘fail to seek help’ as this can imply that non-help seeking may be a failure on the part of the individual and not the system 

2. While the findings are helpful to inform future efforts, it must be noted - in the title, abstract, and limitations - that this was a single-arm study without a comparator, to ensure readers are aware that the results found were from a single arm. 

Thank you and all the best,

Dr. Sandersan Onie

---

## [Editor Report]

Thank you for your revisions to the article. It is an important contribution to the literature. 

All the best,

Dr Sandersan Onie